# Epidemiology, outcomes, and the use of intensive care unit resources of critically ill patients diagnosed with COVID-19 in Sao Paulo, Brazil: A cohort study

Rachel Lane Socolovithc[1], Renata Rego Lins Fumis[1,2]*, Bruno Martins Tomazini[1,2], Laerte Pastore[1], Filomena Regina Barbosa Gomes Galas[3,4], Luciano Cesar Pontes de Azevedo[2,5], Eduardo Leite Vieira Costa[2,6]

1 Adult Intensive Care Unit, Hospital Sírio-Libanês, São Paulo, Brazil, 2 Research and Education Institute, Hospital Sírio-Libanês, São Paulo, Brazil, 3 Cardiologic Intensive Care Unit, Hospital Sírio-Libanês, São Paulo, Brazil, 4 Intensive Care Unit, Heart Institute (Incor), University of São Paulo, São Paulo, Brazil, 5 Emergency Medicine, Hospital das Clínicas, University of São Paulo, São Paulo, Brazil, 6 Pulmonary Division, Cardio-Pulmonary Department, Heart Institute (Incor), University of São Paulo, São Paulo, Brazil

* renata.rlfumis@hsl.org.br

**Data Availability Statement:** All relevant data are within the manuscript and its Supporting information files.

## Abstract

### Background

The coronavirus disease (COVID-19) pandemic has brought significant challenges world-wide, with high mortality, increased use of hospital resources, and the collapse of healthcare systems. We aimed to describe the clinical outcomes of critically ill COVID-19 patients and assess the impact on the use of hospital resources and compare with critically ill medical patients without COVID-19.

### Methods and findings

In this retrospective cohort study, we included patients diagnosed with COVID-19 admitted to a private ICU in Sao Paulo, Brazil from March to June 2020. We compared these patients with those admitted to the unit in the same period of the previous year. A total of 212 consecutive patients with a confirmed diagnosis of COVID-19 were compared with 185 medical patients from the previous year. Patients with COVID-19 were more frequently males (76% vs. 56%, p<0.001) and morbidly obese (7.5% vs. 2.2%, p = 0.027), and had lower SAPS 3 (49.65 (12.19) vs. 55.63 (11.94), p<0.001) and SOFA scores (3.78 (3.53) vs. 4.48 (3.11), p = 0.039). COVID-19 patients had a longer ICU stay (median of 7 vs. 3 days, p<0.001), longer duration of mechanical ventilation (median of 9 vs. 4 days, p = 0.003), and more frequent tracheostomies (10.8 vs. 1.1%, p<0.001). Survival rates until 28 days were not statistically different (91% vs. 85.4%, p = 0.111). After multivariable adjustment for age, gender, SAPS 3, and Charlson Comorbidity Index, COVID-19 remained not associated with survival at 28 days (HR 0.59, 95% CI 0.33–1.06, p = 0.076). Among patients who underwent invasive mechanical ventilation, the observed mortality at 28-days was 16.2% in COVID-19 patients compared to 34.6% in the previous year.

**Funding:** The study was funded by the Sírio-Libanês Hospital. The funders had no role in study design, data collection and analysis, decision to publish, or preparation of the manuscript. The authors received no specific funding for this work.

**Competing interests:** The authors have declared that no competing interests exist.

## Conclusions

COVID-19 required more hospital resources, including invasive and non-invasive ventilation, had a longer duration of mechanical ventilation, and a more prolonged ICU and hospital length of stay. There was no difference in all-cause mortality at 28 and 60 days, suggesting that health systems preparedness be an important determinant of clinical outcomes.

## Introduction

The outbreak of severe acute respiratory syndrome due to a newly identified subtype of coronavirus called SARS-CoV-2 first emerged in Wuhan in December 2019 [1]. The coronavirus disease 2019 (COVID-19) has rapidly spread worldwide [2, 3], leading to the declaration of Public Health Emergency of International Concern by the World Health Organization (WHO) on January 30, 2020 [4].

COVID-19 has a broad spectrum of clinical manifestations from mild nonspecific symptoms such as fever, fatigue, anosmia, cough (productive or not), and gastrointestinal symptoms to severe acute respiratory failure, renal failure, and need of hemodynamic support. Critically ill patients with COVID-19 and acute organ failures require prolonged ICU stay and have a high mortality rate, especially those requiring invasive mechanical ventilation [5–9].

In patients with COVID-19, age has been pointed out as a major risk factor for more severe disease and mortality [10]. Comorbidities are present in more than 30% of cases and are also associated with increased mortality risk [2, 11]. Also, COVID-19 has a higher incidence in men, which are 50% more likely to die from COVID-19 than women [12, 13].

With its rapid spread, COVID-19 created a steep demand for hospital and critical care beds. This increased need for hospital resources led to the collapse of health care systems worldwide, which may have contributed to the higher mortality rates reported [14]. In countries with already overwhelmed health care systems, there were not enough resources from medical equipment to pharmacological drugs and trained personnel to deal with the rising number of patients with COVID-19 in need of hospital support [15].

In late February 2020, the first case of COVID-19 was reported in Brazil. As of June 2020, Brazil had more than 1,300,000 confirmed cases and 57,622 deaths, while worldwide COVID-19 had 503,907 deaths [16]. In this scenario, several Brazilian states have registered a lack of drugs, mechanical ventilators, unavailability of intensive care beds, and the collapse of local health systems [17]. However, the availability of these resources varies between Brazilian regions and between public and private hospitals.

We aimed to describe the clinical characteristics, outcomes, and resource utilization of critically ill patients diagnosed with COVID-19 and assess the impact on the use of hospital resources in comparison with the previous year.

## Methods

### Study design

We performed a retrospective cohort study of patients with COVID-19 admitted in a 32-bed ICU from March to June 2020 in Hospital Sírio-Libanês, São Paulo, Brazil. Originally a mixed surgical-medical intensive care unit (ICU) with daily multidisciplinary rounds, established protocols for patient care and appropriate professional-to-bed ratio, this ICU was dedicated exclusively to the care of COVID-19 patients during the study period. For comparison, we

included medical patients admitted to the ICU due to respiratory or infectious causes during the same months in the previous year.

During the pandemic, the hospital developed a protocol for ICU admission of COVID-19 patients. The main indications for ICU admission were: the need for invasive mechanical ventilation or non-invasive ventilatory support (high-flow nasal cannula and non-invasive positive-pressure ventilation), hemodynamic instability defined as hypotension (mean arterial pressure < 65mmHg) or need of vasopressor support, decreased level of consciousness, and need of renal replacement therapy for acute kidney injury.

### Ethical approval

The ethics committee of the Hospital Sirio-Libanês (approval number 1710) approved the study and waived the need for informed consent. The database was accessed on August 25[th], 2020.

### Patients and data collection

The COVID-19 cohort consisted of all consecutive adult patients admitted to the ICU from March 08[th] to June 30[th], 2020. In 2020, all patients admitted to the ICU had a diagnosis of COVID-19. For the non-COVID-19 cohort, we included all adult patients admitted to the ICU due to respiratory or infectious diseases in the same period in 2019. Patients under 18 years old were excluded in both cohorts.

We used data from an administrative, electronic database of patients admitted to the ICU (Epimed Solutions®, Rio de Janeiro, Brazil), which collects demographic (age, gender and comorbidities), admission (diagnosis, presence of infection), resource utilization (mechanical ventilation, renal replacement therapy, mechanical ventilation, transfusion, type of nutrition), clinical (laboratory, antibiotic use), severity scores and outcomes (length of stay and mortality). A dedicated case manager routinely entered all consecutive cases in the database obtaining information from the hospital's electronic medical record and directly from ICU physicians.

We retrieved data on demographic and clinical characteristics, Simplified Acute Physiology Score (SAPS) 3 (the SAPS 3 score is calculated from 20 variables at the ICU admission of the patient and ranges from 0 to 217, with higher scores indicating a higher risk of death, Sequential Organ Failure Assessment (SOFA) score (the SOFA score is measured in 6 organ systems (cardiovascular, hematologic, gastrointestinal, renal, pulmonary and neurologic), with each organ scoring from 0 to 4, resulting in an aggregated score that ranges from 0 to 24, with higher scores indicating greater dysfunction) on the first day of ICU admission, resources utilization (Yes/No) in the ICU such as mechanical ventilation, transfusion, renal replacement therapy, vasopressors use, and extracorporeal membrane oxygenation (ECMO), as well as the clinical outcomes of all-cause 28 and 60 days survival rate, ICU and hospital length of stay (LOS), and duration of mechanical ventilation (MV).

### Statistical analysis

Comparisons of proportions were performed using chi-square tests for equal proportion or Fisher exact tests where appropriate. Continuous variables were compared using Student t-tests and presented as means (SDs) or were tested using Wilcoxon rank-sum tests and presented as median (interquartile range [IQR]) when appropriate.

We had complete data for the outcome of all-cause mortality at 28 days. For the endpoint of all-cause mortality at 60 days, we censored inpatients with less than 60 days follow-up. We also compared survival curves limiting the analyses to patients who underwent mechanical ventilation. COVID-19 was the primary exposure variable in the time-to-event analyses. We built

Kaplan-Meier curves and applied log-rank tests. We used Cox proportional hazard regression for multivariable adjustment for the variables significantly associated with COVID-19. For this analysis, we reported hazard ratios (HR) and 95% confidence intervals (CI). A two-sided P value of 0.05 was considered statistically significant. Analyses were performed using R software (R Core Team, 2016, Vienna, Austria).

## Results

A total of 575 medical patients were admitted to the ICU from March to June in 2019 and 2020. Of these, we excluded 178 patients admitted due to causes other than respiratory or infectious. In 2020, 212 patients were admitted with a diagnosis of COVID-19. Table 1

**Table 1. Patients characteristics.**

|  | Non-COVID-19 | COVID-19 | p-Value |
|---|---|---|---|
|  | n = 185 | n = 212 |  |
| **Age—years** | 72.36 (17.34) | 65.19 (16.29) | <0.001 |
| <30 | 5 (2.7) | 2 (0.9) |  |
| 30–40 | 8 (4.3) | 15 (7.1) |  |
| 40–50 | 10 (5.4) | 21 (9.9) |  |
| 50–60 | 15 (8.1) | 39 (18.4) |  |
| 60–70 | 25 (13.5) | 46 (21.7) |  |
| 70–80 | 39 (21.1) | 50 (23.6) |  |
| 80–90 | 59 (31.9) | 26 (12.3) |  |
| >90 | 24 (13.0) | 13 (6.1) |  |
| **Gender** |  |  |  |
| Male | 103 (55.7) | 161 (75.9) | <0.001 |
| **Comorbidities** |  |  |  |
| Systemic Arterial Hypertension | 93 (50.3) | 112 (52.8) | 0.683 |
| Diabetes | 50 (27.0) | 54 (25.5) | 0.813 |
| Morbid Obesity | 4 (2.2) | 16 (7.5) | 0.027 |
| Chronic Renal Failure | 29 (15.7) | 17 (8.0) | 0.026 |
| Dyslipidemia | 43 (23.2) | 59 (27.8) | 0.353 |
| Coronary Heart Disease | 29 (15.7) | 37 (17.5) | 0.734 |
| Hypothyroidism | 42 (22.7) | 35 (16.5) | 0.153 |
| Immunosuppression | 31 (16.8) | 16 (7.5) | 0.007 |
| Hematologic Malignancy | 17 (9.2) | 6 (2.8) | 0.013 |
| Solid Tumor | 42 (22.7) | 17 (8.0) | <0.001 |
| COPD | 11 (5.9) | 3 (1.4) | 0.030 |
| Asthma | 4 (2.2) | 6 (2.8) | 0.918 |
| Alcoholism | 3 (1.6) | 4 (1.9) | 1.000 |
| **Charlson Comorbidity Score** |  |  | <0.001 |
| 0 | 53 (28.6) | 114 (53.8) |  |
| 1–3 | 68 (36.8) | 70 (33.0) |  |
| 3–11 | 64 (34.6) | 28 (13.2) |  |
| **SAPS-3** | 55.63 (11.94) | 49.65 (12.19) | <0.001 |
| **SOFA score on day 1** | 4.48 (3.11) | 3.78 (3.53) | 0.039 |

Data are presented as mean (SD) or frequency (proportions).

COPD: Chronic Obstructive Pulmonary Disease, SAPS-3: Simplified Acute Physiology Score, SOFA: Sequential Organ Failure Assessment.

summarizes the baseline characteristics of 212 patients with COVID-19 and 185 patients without COVID-19 included in 2019. In COVID-19 patients, the most prevalent age group was between 60–80 years (total of 45.3%), with a mean age of 65.2 (16) years, on average seven years younger than patients from 2019 (Table 1). Mortality according to age category in both ventilated and non-ventilated patients is shown in Fig 1A and 1B. Non-COVID-19 patients from 2019 had more severe illness when compared to COVID-19 patients, as shown by the SAPS 3 and SOFA scores (Table 1). We observed a higher proportion of males (55.7% vs. 75.9%, p < 0.001) and of morbid obesity (7.5% versus 2.2%, p = 0.027) in COVID-19 patients as compared to 2019 controls. There was no significant difference in the prevalence of comorbidities such as arterial hypertension, diabetes mellitus, and dyslipidemia between the cohorts. Conversely, chronic renal failure, chronic obstructive pulmonary disease, and cancer were more prevalent in the 2019 cohort (Table 1).

The use of critical care resources was markedly higher in COVID patients (Table 2) as compared to non-COVID-19 patients. Invasive mechanical ventilation was more than three times as frequent, and renal replacement therapy and ECMO were more often required. The duration of mechanical ventilation was five days longer, with a median of 9 [IQR 6, 16] days versus 4 [IQR 2, 12] days. Consequently, hospital and ICU lengths of stay increased. ICU stay went from a median of 3.0 [IQR 2.0, 4.0] days in 2019 to 7.0 [IQR 2.0, 15.0] days in 2020 (Table 2). Survival was similar between COVID-19 and non-COVID-19 patients, with 85.4% of the patients alive in 28 days in the 2019 cohort as compared to 91% of the COVID-19 patients in 2020, p-Value = 0.068 (Fig 2). After multivariable adjustment for age, gender, SAPS 3, and Charlson Comorbidity Index, COVID-19 remained not associated with survival at 28 days (HR 0.55, 95% CI 0.28–1.08, p = 0.083). Mortality at 60 days was 17.3% in 2019 and 10.8% in 2020 (Fig 3).

Among patients who underwent invasive mechanical ventilation, the observed mortality at 28-days was 16.2% in COVID-19 patients compared to 34.6% in non-COVID patients (Fig 4), and the mortality at 60-days was 19.0% versus 42.3%, respectively, p-Value (Fig 5).

## Discussion

We herein reported the clinical characteristics and outcomes of 212 patients with COVID-19 admitted to the ICU of a private hospital in Sao Paulo, Brazil, from March to June 2020. We compared their use of hospital resources to 185 historical controls from the previous year. The majority of patients were older men with a past medical history of hypertension and diabetes. When compared to historical controls, critically ill patients with COVID-19 required more invasive and non-invasive ventilatory support, had a longer duration of mechanical ventilation, and a more prolonged ICU and hospital length of stay. There was no difference in all-cause mortality at 28 and 60 days.

Patients in our cohort had a mean age of 65 years and had diabetes and hypertension as the most common comorbidities. These findings are compatible with other international cohorts [5, 18–20]. Data collected by the Brazillian Association of Critical Care [21] in the same period showed that the mean age of ICU patients in Brazilian private hospitals was 60.8 years. These findings likely reflect the fact that older age, hypertension, and diabetes are prevalent conditions [22] and are suggestive that this population might be more susceptible to becoming critically ill, irrespective of the etiology of the acute disease.

We found a low mortality rate in COVID-19 patients compared to average COVID-19 mortality of patients admitted to the ICU [23, 24], but comparable to reports from Asia [25], Europe [26], and North America [20, 27]. Of note, the mortality of COVID-19 patients was not different from the mortality of non-COVID-19 patients in 2019 and was compatible with

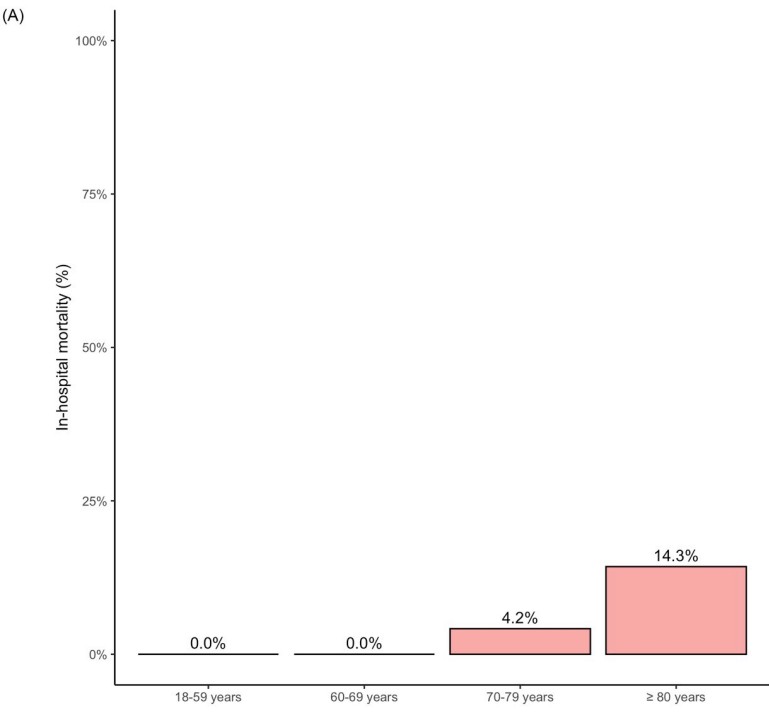

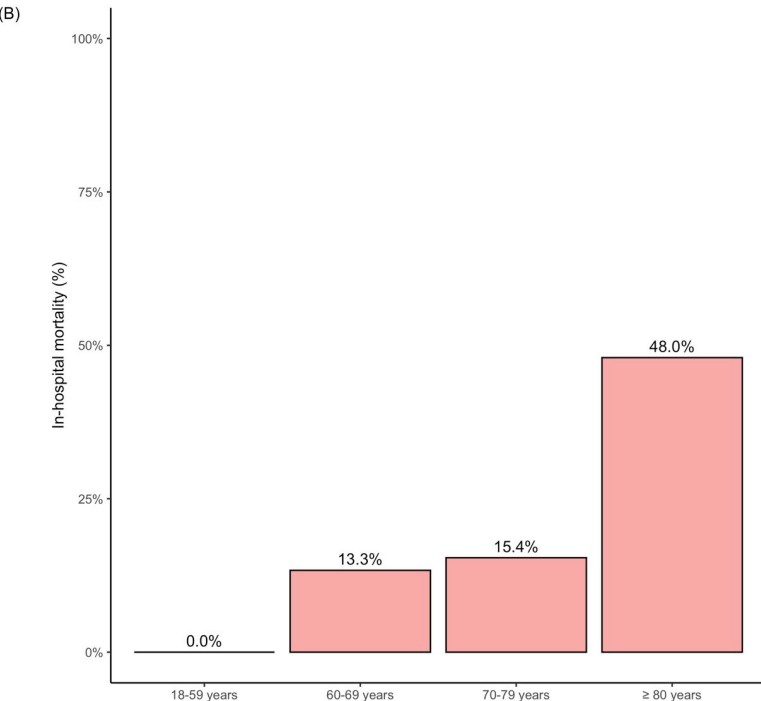

**Fig 1.** A. In-hospital mortality by age category in non-ventilated COVID-19 patients. B. In-hospital mortality by age category in ventilated COVID-19 patients.

**Table 2. Use of hospital resources and outcomes.**

| | Non-COVID-19 | COVID-19 | p-Value |
|---|---|---|---|
| | n = 185 | n = 212 | |
| **Invasive support** | | | |
| Vasopressors–n (%) | 84 (45.4) | 108 (50.9) | 0.317 |
| Invasive mechanical ventilation–n (%) | 26 (14.1) | 105 (49.5) | <0.001 |
| Renal replacement therapy–n (%) | 9 (4.9) | 28 (13.2) | 0.007 |
| ECMO–n (%) | 0 (0.0) | 8 (3.8) | 0.021 |
| Transfusion–n (%) | 24 (13.0) | 35 (16.5) | 0.397 |
| Parenteral Nutrition–n (%) | 5 (2.7) | 13 (6.1) | 0.163 |
| Non-invasive positive pressure ventilation–n (%) | 55 (29.7) | 104 (49.1) | <0.001 |
| High flow nasal cannula–n (%) | 12 (6.5) | 99 (46.7) | <0.001 |
| Tracheostomy–n (%) | 2 (1.1) | 23 (10.8) | <0.001 |
| **Outcomes** | | | |
| Duration of MV days–median [IQR] | 4.00 [2.00–11.75] | 9.00 [6.00–16.00] | 0.003 |
| NIV failure–n (%) | 4 (2.2) | 43 (20.3) | <0.001 |
| 28-day all-cause mortality–n (%) | 27 (14.6) | 19 (9.0) | 0.066 |
| 60-day all-cause mortality–n (%) * | 32 (17.3) | 23 (10.8) | 0.087 |
| 28-day mortality in ventilated patients–n (%) * | 9/26 (34.6%) | 17/105 (16.2%) | 0.114 |
| ICU LOS–days median [IQR] | 3.00 [2.00–4.00] | 7.00 [2.00–15.00] | <0.001 |
| Hospital LOS–days median [IQR] | 12.00 [7.00–24.00] | 17.50 [11.00–31.00] | <0.001 |

*Three patients were excluded because they were still in patients with a follow-up shorter than 60 days.

their predicted mortality. Considering their mean SAPS 3 score, the 60-day mortality rate in COVID-19 patients in our cohort (11.3%) was in accordance with their in-hospital predicted mortality (11.8%), although lower than the predicted mortality for South America (22.4%). The same was true for non-COVID-19 patients from 2019, with 60-day mortality of 17.3% in our cohort and a predicted mortality according to SAPS 3 of 20.3% (and 35.5% for South America). One possible explanation for our lower-than-average mortality in comparison to other COVID-19 cohorts is that we had time to prepare for the pandemic with more than two months of head-start over Asia and Europe. We took the time to learn from their experience, to adjust institutional protocols, and allocate resources accordingly. As a result, we were never in shortage of human resources or medical equipment. For example, scheduled surgeries were canceled, and entire ICUs and floors were reserved to treat exclusively COVID-19 patients, even when cases were only starting in Brazil back in March 2020. The plan ensured that every patient in need of an ICU bed would promptly be admitted to the ICU. All ICU shifts were covered by at least three board certified intensivists, and adequate nurse and respiratory-threapist-to-bed-ratio. We speculate whether the increased mortality published in some COVID-19 series could be attributed to saturation of the health care systems and trained professionals rather than to intrinsic characteristics of the infection.

To our knowledge, this is the first report from South America with demographics, clinical outcomes, and ICU resources used, comparing the impact of the COVID-19 in the ICU to historical controls in the same period of the previous year. Another strength of the study is that we had complete 28-day follow-up of all 212 patients and 60-day follow-up of 209/212 (98.5%)

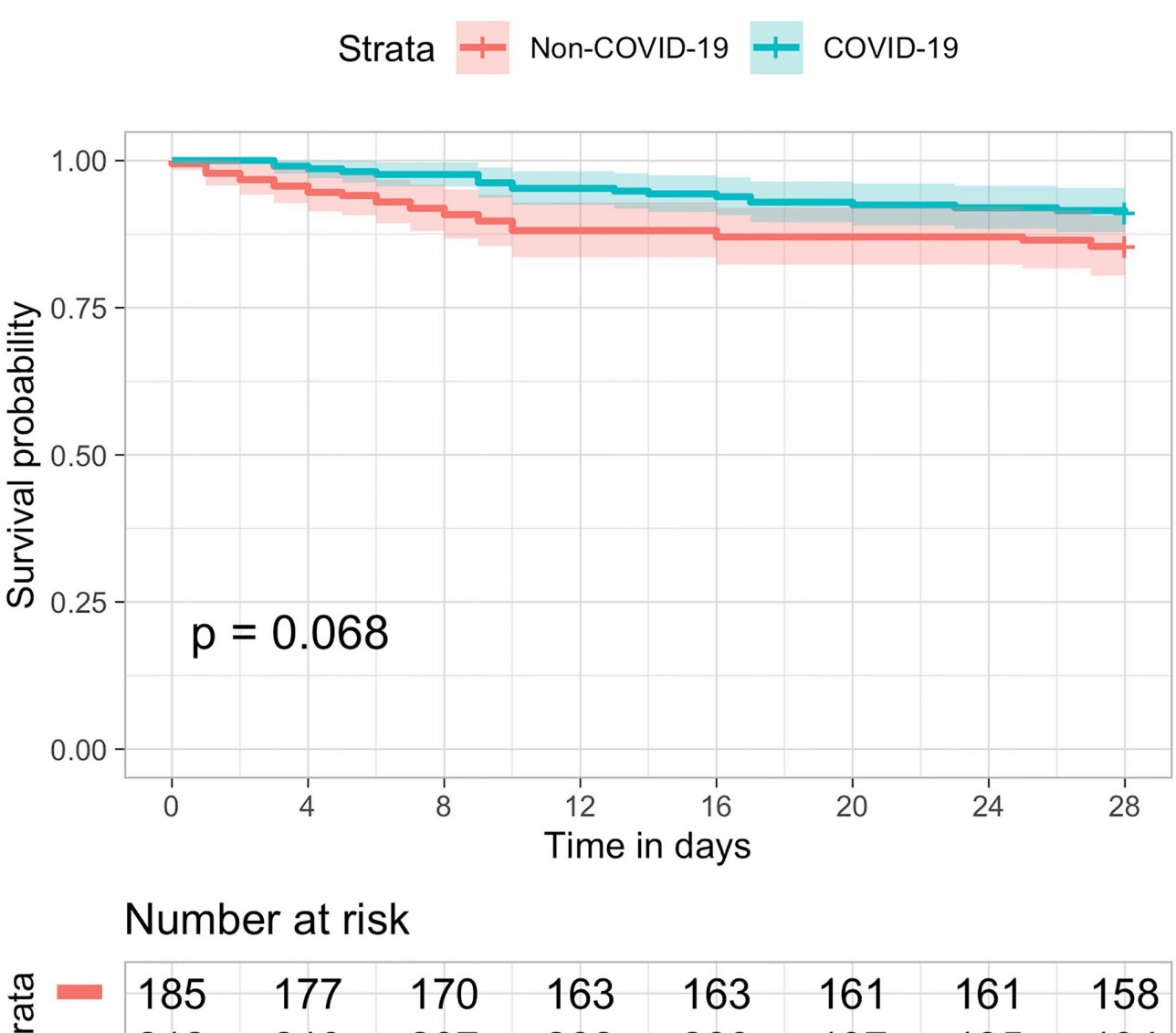

**Fig 2. Kaplan–Meier estimates of all-cause survival rate up to 28 days.** Symbols (tick marks) indicate censored data. Overall survival was not significant different in COVID-19 (blue) as compared to non-COVID patients (orange) in the previous year (logrank p = 0.068).

patients This study, however, has several limitations. First, it is a single-center study performed in a private hospital in São Paulo. While we do not think our results generalize to the public healthcare system, they most likely can be extrapolated to some of the private hospitals in Brazil which account for more than half of ICU beds in the country [28] or even to hospitals in developed countries. Second, we did not have enough non-COVID-19 viral pneumonias to

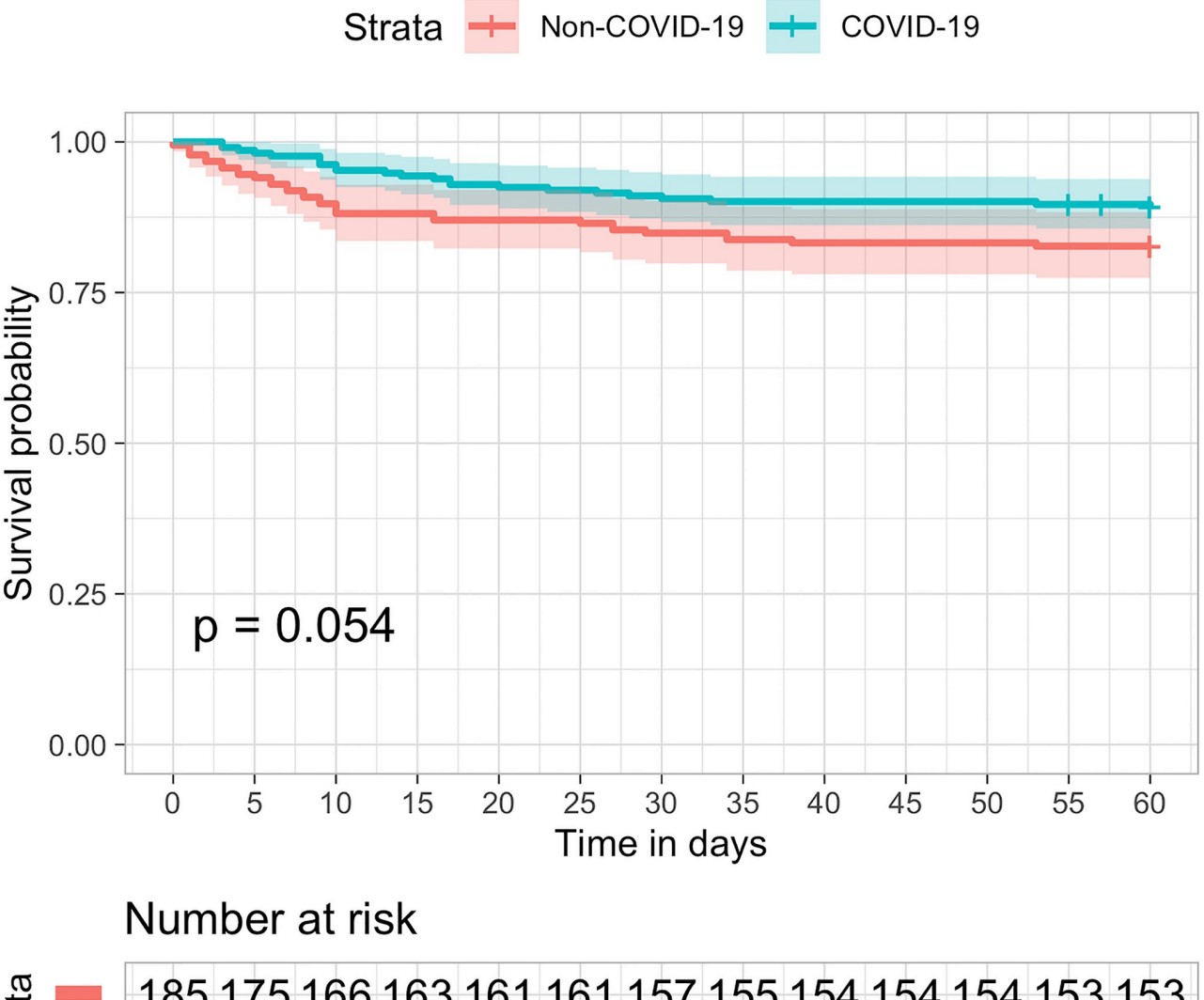

**Fig 3. Kaplan–Meier estimates of all-cause survival rate up to 60 days.** Symbols (tick marks) indicate censored data. Overall survival was not significant different in COVID-19 (blue) as compared to non-COVID patients (orange) in the previous year (logrank p = 0.054).

use as a control group. Instead, we included patients from 2019 with respiratory and infectious causes of ICU admission. Third, this was a retrospective study, with data collected from an administrative database. Finally, we did not access complementary therapies offered to patients, such as glucocorticoids, antivirals, anticoagulation, convalescent plasma, and others.

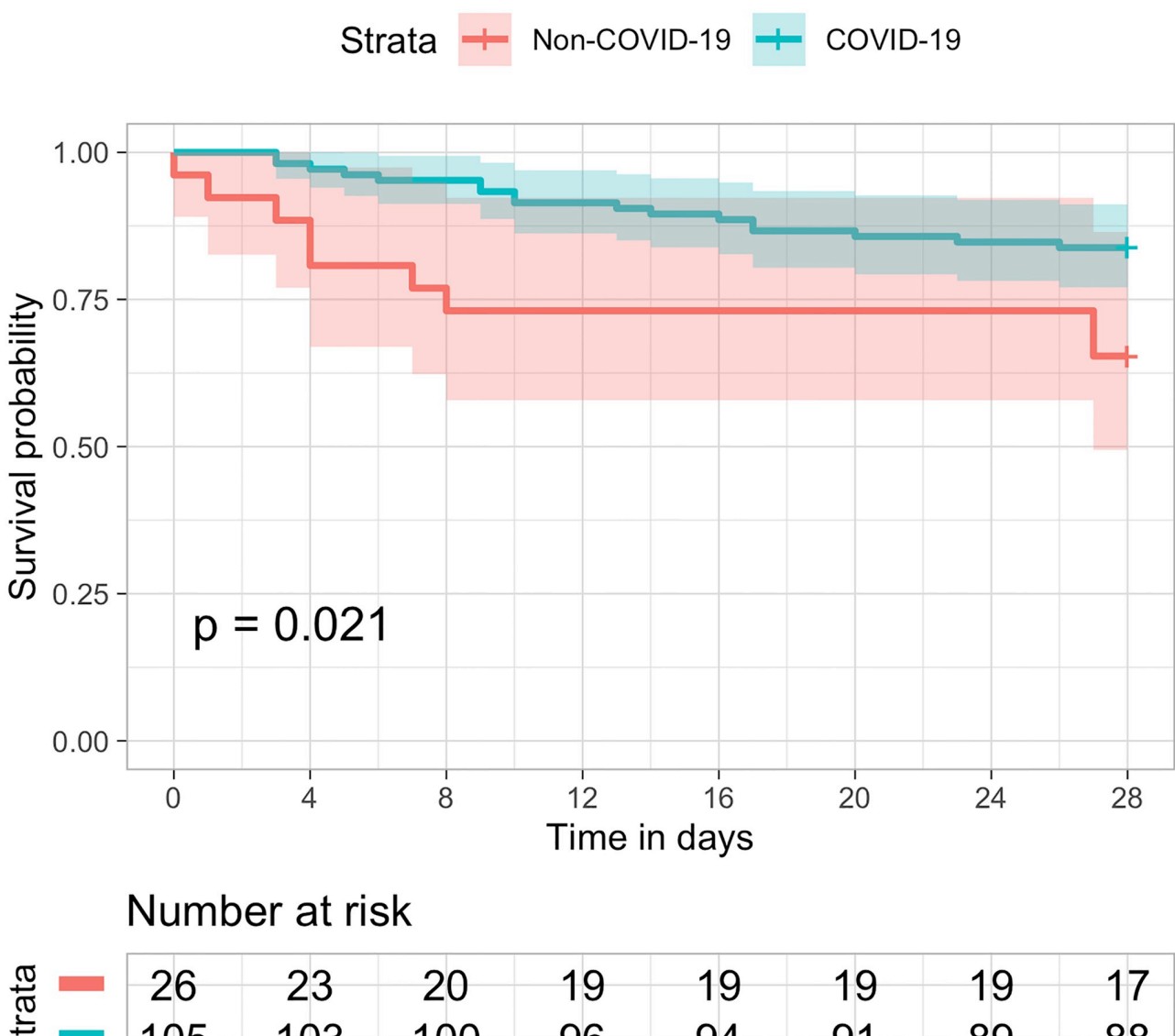

**Fig 4. Kaplan–Meier estimates of all-cause survival rate up to 28 days in ventilated patients.** Overall survival was longer in COVID-19 (blue) as compared to non-COVID patients (orange) in the previous year (logrank p = 0.021).

## Conclusion

COVID-19 required more hospital resources, including invasive and non-invasive ventilation, had a longer duration of mechanical ventilation, and a more prolonged ICU and hospital length of stay. There was no difference in all-cause mortality at 28 and 60 days, suggesting that health systems preparedness be an important determinant of clinical outcomes.

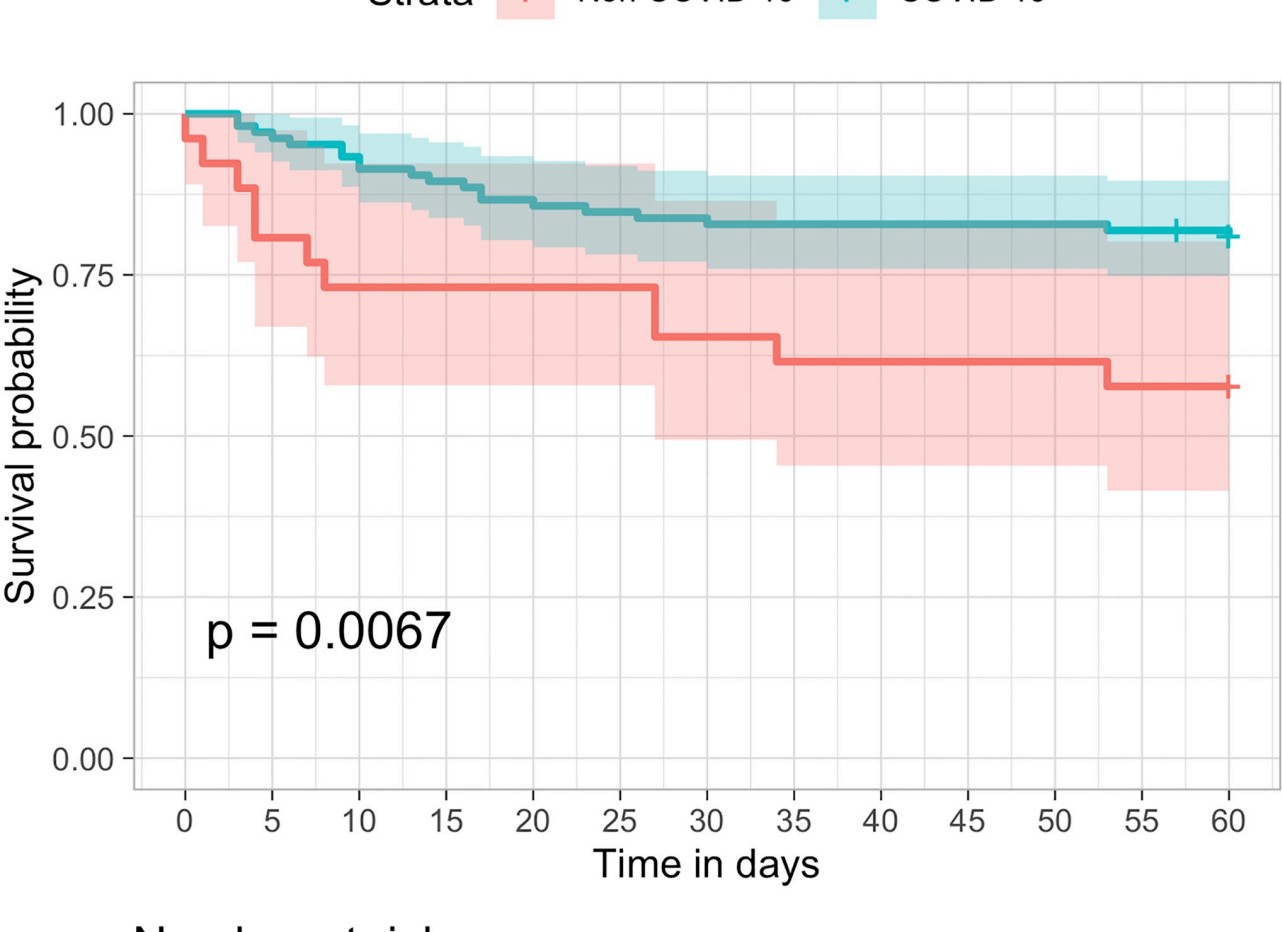

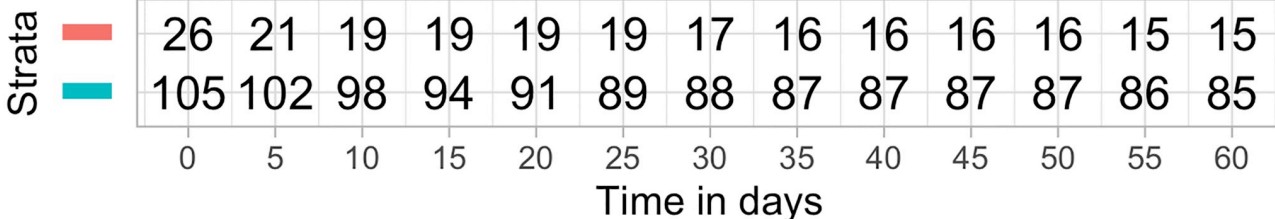

**Fig 5. Kaplan–Meier estimates of all-cause survival rate up to 60 days in ventilated patients.** Overall survival was longer in COVID-19 (blue) as compared to non-COVID patients (orange) in the previous year (logrank p = 0.0067).

## Supporting information

**S1 Data.**
(CSV)

## Acknowledgments

We are grateful to all patients who were admitted to our ICU diagnosed with COVID-19 and their families for believing in us. We also thank the ICU multidisciplinary team for their hard work carried out with competence, resilience, and humanity.

## Author Contributions

**Conceptualization:** Rachel Lane Socolovithc, Renata Rego Lins Fumis, Laerte Pastore, Luciano Cesar Pontes de Azevedo, Eduardo Leite Vieira Costa.

**Data curation:** Rachel Lane Socolovithc, Eduardo Leite Vieira Costa.

**Formal analysis:** Eduardo Leite Vieira Costa.

**Investigation:** Rachel Lane Socolovithc, Renata Rego Lins Fumis, Eduardo Leite Vieira Costa.

**Methodology:** Rachel Lane Socolovithc, Renata Rego Lins Fumis, Bruno Martins Tomazini, Eduardo Leite Vieira Costa.

**Project administration:** Renata Rego Lins Fumis.

**Supervision:** Eduardo Leite Vieira Costa.

**Validation:** Bruno Martins Tomazini, Eduardo Leite Vieira Costa.

**Visualization:** Bruno Martins Tomazini, Eduardo Leite Vieira Costa.

**Writing – original draft:** Rachel Lane Socolovithc, Renata Rego Lins Fumis, Bruno Martins Tomazini, Laerte Pastore, Filomena Regina Barbosa Gomes Galas, Luciano Cesar Pontes de Azevedo, Eduardo Leite Vieira Costa.

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
