## [Decision Letter · Decision Letter 0]

28 Oct 2020

PONE-D-20-28770

Epidemiology, outcomes, and the use of intensive care unit resources of critically ill patients diagnosed with COVID-19 in Sao Paulo, Brazil: a cohort study

PLOS ONE

Dear Dr. Fumis,

Thank you for submitting your manuscript to PLOS ONE. After careful consideration, we feel that it has merit but does not fully meet PLOS ONE’s publication criteria as it currently stands. Therefore, we invite you to submit a revised version of the manuscript that addresses the points raised during the review process.

This is an accurate description of the epidemiologic characteristis of critically ill COVID patients in Brazil. We suggest the Authors to better clarify, in the discussion section, the peculiarity of their population and the clinical impact of their findings. 

We look forward to receiving your revised manuscript.

Kind regards,

Chiara Lazzeri

Academic Editor

PLOS ONE

Journal Requirements:

2.Thank you for including your ethics statement:  "The local ethics committee (approval number 1710) approved the study and waived the need for informed consent.".   

SEH (Associate Editor) 09 16 2020: ***Ed Office, at PRTC, please send back with the following note. At RTC, please check the authors' response and ping me for follow-up if the authors don't address this:

"Please include the date(s) on which you accessed the databases or records to obtain the data used in your study."

3.Thank you for stating the following financial disclosure:

 [NO].

4.We note that you have indicated that data from this study are available upon request. PLOS only allows data to be available upon request if there are legal or ethical restrictions on sharing data publicly. For information on unacceptable data access restrictions, please see http://journals.plos.org/plosone/s/data-availability#loc-unacceptable-data-access-restrictions.

Reviewers' comments:

Reviewer's Responses to Questions

**Comments to the Author**

1. Is the manuscript technically sound, and do the data support the conclusions?

Reviewer #1: Yes

2. Has the statistical analysis been performed appropriately and rigorously? 

Reviewer #1: Yes

3. Have the authors made all data underlying the findings in their manuscript fully available?

Reviewer #1: No

4. Is the manuscript presented in an intelligible fashion and written in standard English?

Reviewer #1: Yes

5. Review Comments to the Author

Reviewer #1: Review comments

Title: Epidemiology, outcomes, and the use of intensive care unit resources of critically ill patients diagnosed with COVID-19 in Sao Paulo, Brazil: a cohort study

General comments

This is a study of relevance for the global community considering the fact that the health outcomes and implications of the COVID-19 pandemic is widespread.

The authors output is satisfactory. The following issues however need to be addressed by the authors:

Title

Epidemiology, outcomes, and the use of intensive care unit resources of critically ill patients diagnosed with COVID-19 in Sao Paulo, Brazil: a cohort study

The letter “a” after the colon should be capitalized.

Abstract

The aim of the study focused on the describing clinical outcomes of patients with COVID-19 yet the methods included another group whose clinical outcomes were also described and compared with the COVID 19 patients.

The conclusion section did not provide the implications of the findings of the study.

Introduction

Line 84 should be more specific to the COVID-19 disease and the appropriate references should be cited. The statement preceding the aim of study suggest a Brazilian regional based challenge which would have required an aim trying to address these regional variations. Therefore, a preceding statement linking the background to the aim of study should be more specific to COVID-19-related critical illness and resources/facilities in ICUs

Methods

1. The methods should be sectioned for better appreciation, i.e. design, study site, participants etc etc

2. What were the attendance records like?

a. how many participants per cohort?

b. Were all patients during the two periods included in the study or there was some form of sampling?

c. What were the inclusion and exclusion criteria?

3. Data collection tools - A detailed description of each of the study tool is required. For example, 1) what are some of the items in each tool, 2) what kind of information do they seek to obtain, 3) how are they scored? 4) how are the scores rated/categorized?

Results

Results reporting percentages should include the n values. For example, n (%). All ‘N’ in tables that represent each cohort should be replaced with ‘n’.

Conclusion

The authors after stating the main findings should provide the implications of their findings to clinical care and the current COVID-19 pandemic.

6. PLOS authors have the option to publish the peer review history of their article (what does this mean?). If published, this will include your full peer review and any attached files.

Reviewer #1: No

---

## [Author Response · Author response to Decision Letter 0]

12 Nov 2020

Reviewer #1: Review comments

Title: Epidemiology, outcomes, and the use of intensive care unit resources of critically ill patients diagnosed with COVID-19 in Sao Paulo, Brazil: a cohort study

General comments

This is a study of relevance for the global community considering the fact that the health outcomes and implications of the COVID-19 pandemic is widespread.

The authors output is satisfactory. The following issues however need to be addressed by the authors:

Title

Epidemiology, outcomes, and the use of intensive care unit resources of critically ill patients diagnosed with COVID-19 in Sao Paulo, Brazil: a cohort study

The letter “a” after the colon should be capitalized.

Authors response: The letter a was capitalized as suggested.

Abstract

The aim of the study focused on the describing clinical outcomes of patients with COVID-19 yet the methods included another group whose clinical outcomes were also described and compared with the COVID 19 patients.

Authors response: Thank you for your comment. The reviewer is correct that the aim was primarily to describe clinical characteristics and outcomes in patients with COVID-19. A secondary aim was to assess the impact of the pandemic on the use of hospital resources. To this end, we chose to compare COVID-19 patients with critically-ill medical patients from the previous year. We are sorry that this was not clear enough in the previous version of the manuscript.

The text now reads: “…and assess the impact on the use of hospital resources in comparison to the previous year”, lines 104-105 of the clean version of the revised manuscript.

The conclusion section did not provide the implications of the findings of the study.

Authors response: The main findings were that COVID-19 patients demanded more hospital resources but had similar clinical outcomes as compared to non-COVID-19 patients. The implications of these findings are that, even in the face of a pandemic, adaptative structural changes and preparedness might influence the outcomes. The text now reads: “There was no difference in all-cause mortality at 28 and 60 days, suggesting that health systems preparedness be an important determinant of clinical outcomes”, lines 51-53 of the clean version of the revised manuscript.

Introduction

Line 84 should be more specific to the COVID-19 disease and the appropriate references should be cited. 

Authors response: We updated the references as suggested. Line 89-90 of the clean version of the revised manuscript.

The statement preceding the aim of study suggest a Brazilian regional based challenge which would have required an aim trying to address these regional variations. Therefore, a preceding statement linking the background to the aim of study should be more specific to COVID-19-related critical illness and resources/facilities in ICUs

Methods

1. The methods should be sectioned for better appreciation, i.e. design, study site, participants etc etc

Authors response: We performed the proposed changes.

2. What were the attendance records like?

a. how many participants per cohort?

Authors response: We included 185 patients in the non-COVID-19 cohort and 212 in the COVID-19 cohort. This information is in the first paragraph of the results section and also in Table 1. Lines 173-174 of the clean version of the revised manuscript.

b. Were all patients during the two periods included in the study or was there some form of sampling?

Authors response: All consecutive adult COVID-19 patients within the study period were included. For the non-COVID-19 cohort, consecutive adult patients admitted to the ICU due to respiratory or infectious diseases were included. The text now reads: “The COVID-19 cohort consisted of all consecutive adult patients admitted to the ICU from March 08th to June 30th, 2020. In 2020, all patients admitted to the ICU had a diagnosis of COVID-19. For the non-COVID-19 cohort, we included all adult patients admitted to the ICU due to respiratory or infectious diseases in the same period in 2019.” Lines 128-131 of the clean version of the revised manuscript.

c. What were the inclusion and exclusion criteria?

Authors response: In the COVID-19 cohort, we included all adult patients admitted within the study time frame. For the non-COVID-19 cohort, we included all adult patients admitted with respiratory or infectious diseases diagnosis from the same period of time in the previous year (2019). Patients under 18 years old were excluded in both cohorts. We added this information in line 132 of the clean version of the revised manuscript.

3. Data collection tools - A detailed description of each of the study tool is required. For example, 1) what are some of the items in each tool, 2) what kind of information do they seek to obtain, 3) how are they scored? 4) how are the scores rated/categorized?

Authors response: We performed the requested changes. Line 141-152 of the clean version of the revised manuscript.

Results

Results reporting percentages should include the n values. For example, n (%). All ‘N’ in tables that represent each cohort should be replaced with ‘n’.

Authors response: We performed the requested changes.

Conclusion

The authors after stating the main findings should provide the implications of their findings to clinical care and the current COVID-19 pandemic.

Authors response: We thank the reviewer for this remark. We performed the changes as suggested. The text now reads: “COVID-19 required more hospital resources, including invasive and non-invasive ventilation, had a longer duration of mechanical ventilation, and a more prolonged ICU and hospital length of stay. There was no difference in all-cause mortality at 28 and 60 days, suggesting that health systems preparedness be an important determinant of clinical outcomes”. Line 286-290 of the clean version of the revised manuscript.

.

---

## [Editor Report · Decision Letter 1]

19 Nov 2020

Epidemiology, outcomes, and the use of intensive care unit resources of critically ill patients diagnosed with COVID-19 in Sao Paulo, Brazil: A cohort study

PONE-D-20-28770R1

Dear Dr. Fumis,

We’re pleased to inform you that your manuscript has been judged scientifically suitable for publication and will be formally accepted for publication once it meets all outstanding technical requirements.

Kind regards,

Chiara Lazzeri

Academic Editor

PLOS ONE
---

## [Editor Report · Acceptance letter]

23 Nov 2020

PONE-D-20-28770R1 

Epidemiology, outcomes, and the use of intensive care unit resources of critically ill patients diagnosed with COVID-19 in Sao Paulo, Brazil: A cohort study 

Dear Dr. Fumis:

I'm pleased to inform you that your manuscript has been deemed suitable for publication in PLOS ONE. Congratulations! Your manuscript is now with our production department. 

Kind regards, 

on behalf of

Dr. Chiara Lazzeri 

Academic Editor

PLOS ONE